# Toward Ultra-Low Efficiency Droop in C-Plane Polar InGaN Light-Emitting Diodes by Reducing Carrier Density with a Wide InGaN Last Quantum Well

**Yongbing Zhao [1] and Panpan Li [2],***

1    School of Physics and Electronics, Yancheng Teachers University, Yancheng 224007, China
2    Department of Electrical and Computer Engineering, University of California, Santa Barbara, CA 93117, USA
*    Correspondence: lipanpannu@gmail.com; Tel.: +1-805-708-6766

**Abstract:** We demonstrate an ultra-low efficiency droop in c-plane polar InGaN blue light-emitting diodes (LEDs) by reducing the carrier density using a wide InGaN last quantum well (LQW). It is found that the LEDs with a 5.2 nm thick LQW show a negligible efficiency droop, with an external quantum efficiency (EQE) reducing from a peak value of 38.8% to 36.4% at 100 A/cm$^2$ and the onset-droop current density is raised from 3 A/cm$^2$ to 40 A/cm$^2$ as the LQW thickness increases from 3.0 nm to 5.2 nm. The analysis based on the ABC model indicates that small efficiency droop is caused by the reduced carrier density using a wide LQW. The peak efficiency is reduced with a wide LQW, which is caused by the reduction of the electron-hole wavefunction overlap and the deterioration of the crystal quality of the InGaN layer. This study suggests that the application of the InGaN LEDs with a wide LQW can be a promising and simple remedy for achieving high efficiency at a high current density.

**Keywords:** InGaN; light-emitting diodes; efficiency droop; carrier recombination; carrier localization

## 1. Introduction

InGaN-based light-emitting diodes (LEDs) have been attractive and widely applied in the field of general lighting, backlighting for displays, automotive lighting and so on due to the wide emission spectra range covering near-ultraviolet to red. [1–3] The efficiency droop, which is one of the main challenges of InGaN-based LEDs since internal quantum efficiency (IQE) reduces by increasing the injection currents, is still extensively investigated. [4] Many theories have attempted to explain such observations, including Auger recombination [5–7], electron leakage, [8,9] lack of hole injection, [10,11] and carrier delocalization [12,13].

In order to keep a high IQE at a high injection current, reducing the carrier density in the emitting quantum wells (QWs) is a suggesting primarily remedy to minimize the Auger recombination as well as the carrier overflow loss [14,15]. A maximum quantum efficiency above 200 A/cm$^2$ has been achieved with a 9 nm wider well thickness for blue InGaN/GaN double-heterostructure LED by Gardner et al. [16], however, InGaN/GaN heterostructure LEDs show a low external quantum efficiency (EQE) due to the lack of the electro confinement effect [17–21]. Li et al. showed that the efficiency droop was alleviated by enlarging the QW thickness due to a smaller carrier overflow [17,18]. Nevertheless, this theory is insufficient since too many factors were involved, such as the change of the piezoelectric field, the electro-hole wavefunction overlap and the carrier density. Schubert et al. showed that both the electron and hole dwell times can be significantly increased with a wide QW thickness by theoretical calculation, but the effect on the efficiency droop is also unclear [19]. Pan et al. realized InGaN blue LEDs with a high quantum efficiency and low-efficiency droop at a

high current density of 400 A/cm$^2$ on a semipolar (20-2-1) plane by employing a thick well of 12 nm, which was due to the low operation carrier density [20]. However, growing InGaN blue LEDs on the semipolar bulk GaN is too expensive and remains impractical for commercialization. Since the majority of carrier recombination happens in the last quantum well (LQW) due to the low mobility of the hole and fast mobility of the electron [21], we investigated the effect of the LQW thickness on the efficiency droop behavior for the c-plane polar InGaN LEDs based on the experimental design and theoretical calculation based on the ABC model.

## 2. Materials and Methods

InGaN-based blue LEDs were realized by the metal-organic chemical vapor deposition (MOCVD) growth and chip process with a mesa structure. Two-inch c-plane (0001) patterned sapphire substrates (PSS) were used for the LEDs epitaxial growth. Trimethylgallium (TMGa), triethylgallium (TEGa), trimethylindium (TMIn), Trimethylaluminum (TMAl), ammonia (NH$_3$), silane (SiH$_4$), and magnesium (Cp$_2$Mg) were used as precursors and dopants. The LEDs schematic structure is shown in Figure 1a, which is composed of a 30 nm low temperature GaN nucleation layer, a 4.0-μm uid_GaN, a 3.0-μm n-GaN, 3 pairs of 2 nm/80 nm In$_{0.04}$Ga$_{0.96}$N/GaN strain released layers, 9 pairs of InGaN/GaN (3 nm/8.5 nm) multiple quantum wells (MQWs), a 20 nm p-Al$_{0.15}$Ga$_{0.85}$N electron blocking layer (EBL), a 100 nm p-GaN main layer and a 15 nm p+GaN contact layer. The thickness of the LQW is designed to be 3.0 nm, 4.7 nm and 5.2 nm, corresponding to an InGaN LQW growth time of 90 s, 135 s and 156 s, respectively. The TEGa and TMIn flow rate for the InGaN QW is set to be $3.6 \times 10^{-4}$ mol/min and $1.3 \times 10^{-4}$ mol/min, respectively. Standard LEDs chips with a size of $1 \times 1$ mm$^2$ were fabricated. Firstly, Indium Tin Oxide (ITO) was deposited on top of the p-type GaN. The n-GaN layer was exposed by inductively coupled plasma etching with a 1.3 μm depth and Ti/Al/Ni/Au metals were deposited as the n-contact layer. Finally, the Cr/Pt/Au metals were deposited as the p/n contact pads. A patterned design of mental electrodes with multiple-branches for both the p/n contact pads was used to reduce the current crowding effect [22]. The devices were diced, packaged with silica gel and measured in an industrial calibrated integrating sphere. Figure 1b demonstrates the electroluminescent (EL) intensity mapping of the InGaN high power LED chip under a driven current of 50 mA. A uniform current spreading can be seen, which is helpful to rule out the effect of current crowding on the efficiency droop.

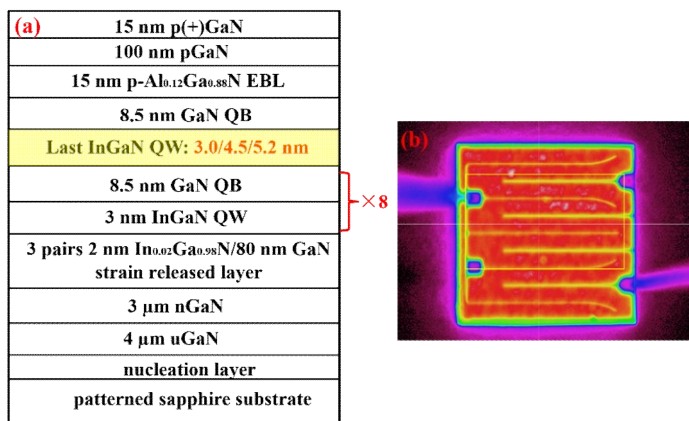

**Figure 1.** (**a**) Schematic epitaxial structure and (**b**) electroluminescent intensity mapping of the InGaN LED chip (1 mm$^2$).

## 3. Results and Discussion

The EL emission peak and the full-width half maximum (FWHM) versus injection current for the three LEDs with different LQWs are shown in Figure 2. It is obvious to see that both the emission wavelength and the FWHM increase with increasing LQW thickness. All the peaks of the three samples present an obvious blue-shift, which is caused by the carriers filling the localized states [23]. The large

blue-shift in the wavelength and wide FWHM for the InGaN LEDs with 5.2-nm LQW are mainly caused by a more inhomogeneous indium distribution as well as a larger piezoelectric field within the active region.

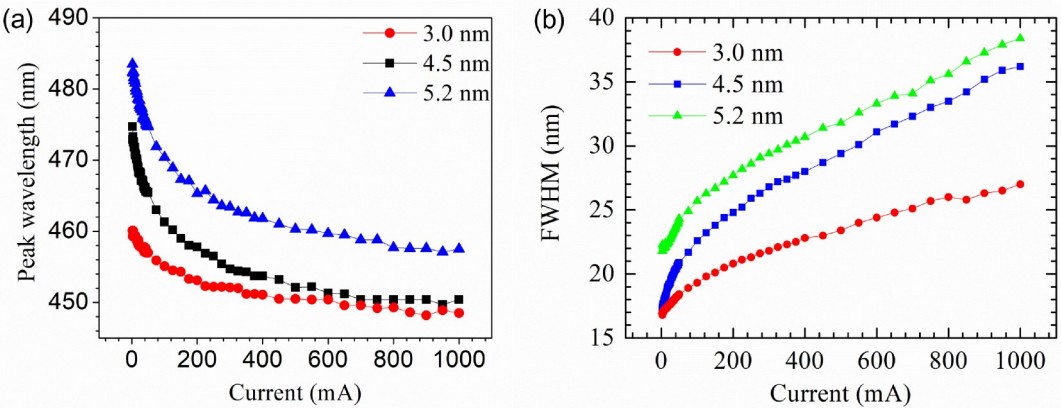

**Figure 2.** (**a**) The electroluminescent (EL) emission peak and (**b**) full-width half maximum (FWHM) versus injection current for the LEDs with different last quantum wells (LQWs) of 3.0 nm, 4.5 nm and 5.2 nm.

The EQE can be calculated by EQE = $(P_{output}/hv)/(I/q)$, where $P_{output}$, $hv$ and $I$ are the output power, photonic energy and the injection current, respectively. The measured absolute and normalized EQE of the three LEDs are plotted in Figure 3a,b. It is clear that the LED with a wider LQW thickness presents a small efficiency droop, a larger onset-droop current and a relatively low peak EQE. Importantly, the LED with an LQW thickness of 5.2 nm shows a negligible efficiency droop, in which the EQE reduces from a peak value of 38.8% to 36.4% at 100 A/cm$^2$. Meanwhile, the onset-droop current density is raised from 3 A/cm$^2$ to 40 A/cm$^2$ as the LQW thickness increases from 3.0 nm to 5.2 nm. The droop ratio is defined as (Max_IQE-IQE)/Max_IQE, where the Max_IQE presents the peak IQE. From Figure 3b, we can see that the efficiency droop ratio is significantly alleviated from 30% to 7% at 100 A/cm$^2$ as the LQW thickness increases from 3.0 nm to 5.2 nm. The peak efficiency decreases from 69.6% to 38.8% as the LQW thickness increases from 3.0 to 5.2 nm, which can be explained by the larger quantum confinement stark effect (QCSE) [17,18] and the subsequent reduction of the electron-hole wavefunction overlap. These results suggest that there is a balance to achieve both high efficiency and efficiency droop for the *c*-plane InGaN LEDs by tuning the LQW thickness, which is one of the easiest and controllable parameters during MOCVD epitaxial growth. It is worth pointing out that the efficiency of LEDs with a wide LQW thickness of 5.2 nm is estimated to surpass the efficiency of the conventional highly efficient LEDs with 3.0-nm thick LQW at a current density of 200 A/cm$^2$ (2A). Therefore, the c-plane polar LEDs with a wide LQW show benefits in efficiency by operating at a high current density due to the ultra-low efficiency droop, which is very easy to be adopted in mass production. We do believe that the peak EQE for the LEDs with a wide LQW can be further improved by optimizing the LQW crystal quality such as adopting a lower InGaN QW growth rate or using the AlGaN insertion layer to reduce the Shockley–Read–Hall (SRH) nonradiative recombination.

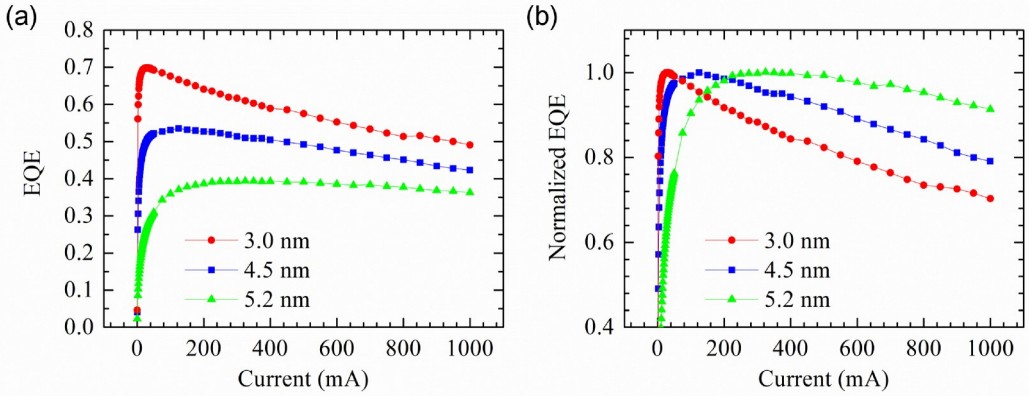

**Figure 3.** (**a**) The measured absolute and (**b**) Normalized external quantum efficiency (EQE) of the LEDs with different LQWs of 3.0 nm, 4.5 nm and 5.2 nm.

A modified ABC model with the consideration of the shrink of InGaN QW is employed in our previous study to investigate the carrier recombination dynamics in InGaN MQWs. IQE can be calculated by [24,25], where $I$ is the current injected into the active region, $V_{\text{active}}$ is the physical volume of the active region, and $q$ is the elementary charge. Due to the localized carriers in the potential minima of In-rich InGaN clusters [26–28], it is reported that the effective active region volume ($V_{\text{effective}}$) is smaller than the $V_{\text{active}}$. Shen et al. showed that the effective QW thickness of InGaN blue LEDs would be reduced to only ~1.0 nm with a $V_{\text{active}}$ of 2.5~3.0 nm [16,29]. Our previous study showed that an effective volume ratio β can be introduced as β = $V_{\text{effective}}/V_{\text{active}}$ in the efficiency equation function to understand the effect of reduced $V_{\text{effective}}$ on the IQE droop due to the InGaN clustering and carrier localization [25].

The equilibrium carrier density $n$ in the QW can be calculated by

$$n = \sqrt{\text{IQE} \cdot I} / \sqrt{BqV_{\text{effective}}} \tag{1}$$

Hence, we can revise it by introducing β

$$n = \sqrt{\text{IQE} \cdot I} / \sqrt{BqV_{\text{active}}\beta} = n\prime / \sqrt{\beta} \tag{2}$$

Which assumes that the carriers have distributed uniformly in the InGaN QW. Therefore, IQE can be calculated by [25]

$$\text{IQE} = \frac{B(n\prime/\sqrt{\beta})^2}{A(n\prime/\sqrt{\beta}) + B(n\prime/\sqrt{\beta})^2 + C(n\prime/\sqrt{\beta})^3} \tag{3}$$

The external quantum efficiency (EQE) is calculated by, where $P_{\text{output}}$ and $h\nu$ are the output power and photonic energy. In our experiment, the light extraction efficiency (LEE) is assumed to be 0.8 for the high-efficiency InGaN LEDs grown on PSS and it has been used in our previous study [24]. Therefore, the IQE can be achieved by considering the LEE as IQE = EQE/LEE.

By fitting the experimental data through Equation (3), accurate-fitting results can be achieved. The fitting IQE versus carrier density for InGaN LEDs is plotted in Figure 4. It is worth pointing out that a large value of the Auger coefficient was not introduced and the extracted Auger coefficient with a value of $2 \times 10^{-30}$ cm$^6$ s$^{-1}$ is more reasonable, which agrees well with the expectations by the experimental measurements and the theoretical calculations [5,6].

The relationship between the carrier density and the driven current with consideration to the reduced effective volumes for the three LEDs are shown in Figure 4a. For a given current, the carrier density of the InGaN LED with a wide InGaN LQW is lower than that of the blue LED. Moreover, the IQE versus carrier density for InGaN LEDs are plotted in Figure 4b. The InGaN LED with 5.2 nm thick LQW has shown a significant alleviation of efficiency droop. Therefore, the larger onset-droop current

and the alleviated droop ratio of LED with a wide InGaN well can be well interpreted by the reduction of the carrier density, which leads to a reduction of Auger recombination and carrier leakage [20].

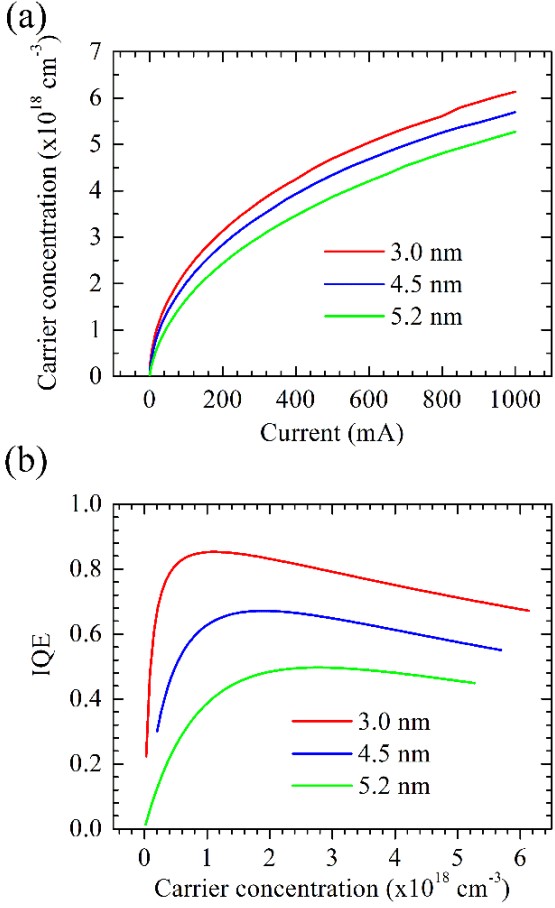

**Figure 4.** (**a**) The relationship between the current and (**b**) internal quantum efficiency (IQE) versus carrier density for the three LEDs with various LQW thicknesses.

## 4. Conclusions

In conclusion, we demonstrate a very simple method to achieve ultra-low efficiency droop for c-plane polar InGaN blue LEDs with a wide LQW. The LEDs with a 5.2-nm thick LQW show a very small efficiency droop with an EQE slightly reducing from a peak of 38.8% to 36.4% at 100 A/cm$^2$. The applications of wide LQW in the commercial polar LEDs are attractive due to the ultra-low efficiency droop, the low cost of the c-plane substrate and their easy adoption in commercialization compared to current approaches like growing small-sized and costly semipolar bulk GaN or using complicated InGaAlN/GaN EBLs designs. We expect the further improvement of the peak EQE to enhance the overall EQE by optimizing the growth condition of LQW and reducing the SRH nonradiative recombination.

**Author Contributions:** P.L. bring the concept and both Y.Z. and P.L. carried out the experiments, results analysis and useful discussion. P.L. wrote the manuscript.

**Funding:** This research was funded by The National Natural Science Foundation of China (11847166) and The Natural Science Foundation of the Jiangsu Higher Education Institutions of China (18KJB510047).

**Conflicts of Interest:** The authors declare no conflict of interest.

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
