# Peer review of "Toward Ultra-Low Efficiency Droop in C-Plane Polar InGaN Light-Emitting Diodes by Reducing Carrier Density with a Wide InGaN Last Quantum Well"

_applsci, doi:10.3390/app9153004_

Round 1

Reviewer 1 Report

This article describes a Toward ultra-low efficiency droop in c-plane polar InGaN light-emitting diodes by reducing carrier density with a wide InGaN last quantum well. The authors claimed that the application of InGaN LEDs with a wide LQW can be a promising and simple remedy for achieving high efficiency at a high current density. I find the manuscript somewhat lacking for publication in applied science. I therefore recommend several additions that the authors may like to consider before the publication is suitable in your Journal.

1.     Page 1, line 38, droop was alleiviated   should be   alleviated

2.     Page 2, line 45, semipolar (20-2-1)?

3.     Page 3, line 46, density.20?

4.     Page 4, line 48, electron,21 ?

5.     Page 3, line 99, “explained by a larger quantum confinement stark effect (QCSE)” reference should be given.

6.     Page 3, line 110, Figure 3b. x-axis Normalized EQE (a.u.), if its normalized what’s meaning of (a.u.)

7.     Page 4, line 119, Shen has shown ? what is Shen; Shen et.al

8.     Page 4, line 135, “By fitting the experimental data through Eq. (3), accurate fitting results can be achieved”. Where is fitting data, The authors should show the fitting curve.

9.     Page 4, line 138, measurements5 ??what’s mean of 5

Author Response

1.     Page 1, line 38, droop was alleiviated   should be   alleviated

Answer: It has been changed in our revised manuscript.

2.     Page 2, line 45, semipolar (20-2-1)?

Answer: Yes, it is semipolar (20-2-1) plane.

3.     Page 3, line 46, density.20?

Answer: It has been changed in our revised manuscript.

4.     Page 4, line 48, electron,21 ?

Answer: It has been changed in our revised manuscript.

5.    Page 3, line 99, “explained by a larger quantum confinement stark effect (QCSE)” reference should be given.

Answer: Reference has been added in our revised manuscript.

6.     Page 3, line 110, Figure 3b. x-axis Normalized EQE (a.u.), if its normalized what’s meaning of (a.u.)

Answer: We have removed (a. u.) in our revised manuscript in Fig. 3.

7.     Page 4, line 119, Shen has shown ? what is Shen; Shen et.al

Answer: We have revised as Shen et al. in reference 5 as shown in Page 4 Line 122.

8.     Page 4, line 135, “By fitting the experimental data through Eq. (3), accurate fitting results can be achieved”. Where is fitting data, The authors should show the fitting curve.

Answer: The fitting IQE versus carrier density for InGaN LEDs are plotted in the Fig. 4, we have added this sentence in Page 4, line 139, which show the IQE versus current carrier concentration based on the EQE versus J in Fig .3(a). Similar report has been published in our previous study of Li, P.; Zhao, Y.; Yi, X.; and Li, H.; Appl. Sciences, 2018, 8, 2138.

9.     Page 4, line 138, measurements5 ??what’s mean of 5

Answer: It is a mistake and we have removed it in our revised manuscript.

Reviewer 2 Report

This manuscript reports on tuning the width of InGaN last quantum well for blue LEDs to reduce the droop in EQE curves in high current range. Blue LEDs with 1 mm2 were fabricated and characterized. The authors observed that with increasing width of quantum wells, the EQE droop was reduced. The study is very interesting and fruitful. It shows merit to improve the design of blue LEDs for industry. Indeed, I have several concerns and would like the authors to make clarifications/revisions.

1. The blue LEDs shown in the manuscript are surface-mounted devices. Note that the current packages for blue LEDs are mostly based on thin-film flip-chip (TFFC) or chip-scale package (CSP) instead of SMD. The current spreading, device architecture, and local current density would vary from case to case. In other words, the optimization approach for InGaN last quantum well might not apply to TFFC or CSP. I would suggest that the authors might want to make this point clear.

2. Please give more details regarding the EQE setup, e.g. instrument.

3. Figure 2 is very hard to read. Please revise.

4. The authors claim that the extraction efficiency is assumes to be 0.8. It is not very clear in the text. One way to estimate extraction efficiency is do low-temperature EQE measurement. How was the value of 0.8 verified?

5. It is not clear how the IQE curves were fitted by the ABC model. Please show the fitting values of A, B, and C. Please compare the values with those in literature.

Author Response

Manuscript ID: applsci-550509

Toward ultra-low efficiency droop in c-plane polar InGaN light-emitting diodes by reducing carrier density with a wide InGaN last quantum well

Jul. 17, 2019

Dear Editor,

Thank you for sending us the reviewers’ comments. We also thank the reviewers for their time and effort in reviewing our work. The suggestions from the reviewers have been fully addressed. All required changes are highlighted in blue.

2. Referee #2, comment:

This manuscript reports on tuning the width of InGaN last quantum well for blue LEDs to reduce the droop in EQE curves in high current range. Blue LEDs with 1 mm2 were fabricated and characterized. The authors observed that with increasing width of quantum wells, the EQE droop was reduced. The study is very interesting and fruitful. It shows merit to improve the design of blue LEDs for industry. Indeed, I have several concerns and would like the authors to make clarifications/revisions.

1. The blue LEDs shown in the manuscript are surface-mounted devices. Note that the current packages for blue LEDs are mostly based on thin-film flip-chip (TFFC) or chip-scale package (CSP) instead of SMD. The current spreading, device architecture, and local current density would vary from case to case. In other words, the optimization approach for InGaN last quantum well might not apply to TFFC or CSP. I would suggest that the authors might want to make this point clear.

Answer: The packaged method we used as shown in the Fig. 1(a), which is typically used to package the LEDs and measure the integrating in the sphere. The TFFC and CSP package are helpful for the current spreading but in our manuscript, we focus the effect epitaxy structure design of last quantum well thickness on the efficiency droop by using standard package.

2. Please give more details regarding the EQE setup, e.g. instrument.

Answer: As we mentioned in Page 2 Line 70: “The devices were diced, packaged with silica gel and measured in an industrial calibrated integrating sphere.” All the LEDs were packaged and messured in the integrating sphere, as shown in the following figures (a) and (b). The EQE can be calculated by EQE = (Poutput/hv)/(I/q), where Poutput, hv and I are output power, photonic energy and the injection current, respectively.

Fig. 1(a) Package LEDs and (b) Integrating sphere to measure the output power.

3. Figure 2 is very hard to read. Please revise.

Answer: It has been revised and divided into two figures and we think it is clear enough now.

3. The authors claim that the extraction efficiency is assumes to be 0.8. It is not very clear in the text. One way to estimate extraction efficiency is do low-temperature EQE measurement. How was the value of 0.8 verified?

Answer: The peak EQE of the blue LEDs with 3.0 nm thick last QW is 70% already, as shown in Fig. 3(a). A light extraction efficiency (LEE) of 0.8 is used then we can estimate the peak IQE of 87%. If a LEE is less than 0.7, the peak IQE would be larger than 1, which is not reasonable. Therefore, we use a LEE of 0.8 and it has been used in our previous paper published (Appl. Sciences, 2018, 8, 2138 and Applied Physics Express 6 (9), 092101, 2013).

5. It is not clear how the IQE curves were fitted by the ABC model. Please show the fitting values of A, B, and C. Please compare the values with those in literature.

Answer: The process to fit EQE~J by ABC model then get IQE~n curve can be shown clearly in our previous papers (Appl. Sciences, 2018, 8, 2138 and Applied Physics Express 6 (9), 092101). First, we need to get the relationship between n and J by . The extracted value of A, B and C for the three samples are summarized in following table. Those values agree what we published in Applied Physics Express 6 (9), 092101, 2013 and the Auger coefficient C is well agreement with the theoretically calculation in Appl. Phys. Lett., 2009, 94, 91109.

LQW thickness

A(s-1)

B(cm3 s-1)

C(cm6 s-1)

3.0 nm

8.82×106

5.66×10-11

2.70×10-30

4.5 nm

3.62×107

3.90×10-11

2.51×10-30

5.2 nm

7.70×107

2.45×10-11

2.00×10-30

Round 2

Reviewer 1 Report

The revised manuscript is well written and organized. Thus I believe that the revised article should be accepted for publication as it is.